# Composite Probiotics Improve Gut Health and Enhance Tryptophan Metabolism in Nursery Piglets During Liquid Feeding

**DOI:** 10.3390/ijms26125698

**Published:** 2025-06-13

**Authors:** Man Du, Qifan Zhang, Yutian Shen, Jie Fu, Yizhen Wang, Bin Yao, Zeqing Lu

**Affiliations:** 1National Engineering Research Center of Green Feeds and Healthy Livestock Industry, Zhejiang University, 866 Yuhang Tang Road, Hangzhou 310058, China; 22017015@zju.edu.cn (M.D.); 12017018@zju.edu.cn (Q.Z.); 22317061@zju.edu.cn (Y.S.); fujie2019@zju.edu.cn (J.F.); yzwang321@zju.edu.cn (Y.W.); 2Key Laboratory of Animal Nutrition and Feed, Ministry of Agricultural and Rural Affairs, Zhejiang University, 866 Yuhang Tang Road, Hangzhou 310058, China; 3Key Laboratory of Molecular Animal Nutrition, Ministry of Education, Zhejiang University, 866 Yuhang Tang Road, Hangzhou 310058, China; 4State Key Laboratory of Animal Nutrition and Feeding, Institute of Animal Science, Chinese Academy of Agricultural Sciences, Beijing 100193, China; binyao@caas.cn

**Keywords:** probiotic, gut microbiota, tryptophan metabolism, liquid feeding

## Abstract

Probiotics are widely used as dietary additives to strengthen gut barrier function, shape microbiota composition, regulate host metabolism, and promote overall health. To enhance probiotic delivery and microbial viability, this study evaluated a liquid feeding system supplemented with a probiotic consortium (*Bifidobacterium infantis*, *Lactobacillus plantarum*, and *Pediococcus acidilactici*) in nursery piglets. A 60-day trial involving 270 piglets (16.84 ± 0.12 kg) compared three diets: solid feed (Dry), liquid feed (Liq), and probiotic-enriched liquid feed (Pro). Compared to the Dry and Liq groups, probiotic supplementation significantly improved growth performance, with the average daily gain increasing by over 17.86% (*p* < 0.01) and the average daily feed intake increasing by more than 6.08% (*p* < 0.05). The feed conversion ratio was reduced by up to 8.08% (*p* < 0.05), indicating improved feed efficiency. The Pro group also exhibited elevated tight junction protein expression (*p* < 0.05), increased colonic short-chain fatty acid levels (*p* < 0.01), and decreased serum biomarkers of intestinal permeability (*p* < 0.05). The 16 S rRNA sequencing indicated the probiotic-driven colonization of *B. infantis* and *L. plantarum* and the suppression of opportunistic pathogens. Metabolomic analyses revealed enhanced colonic tryptophan metabolism, evidenced by elevated kynurenic and xanthurenic acid levels. Additionally, serum-targeted metabolomics and in vitro experiments confirmed that *B. infantis* and *L. plantarum* effectively converted tryptophan into indole-3-lactic acid, promoting its accumulation in piglet serum and colons. These results deepen our understanding of the mechanisms by which probiotics and tryptophan metabolism enhance intestinal health, providing a foundational platform for the application of probiotic-based interventions in livestock production.

## 1. Introduction

Probiotics, characterized as viable microbial preparations that enhance host physiology at appropriate dosages, serve as functional additives within modern dietary regimens and agricultural practices [1]. Extensive evidence has demonstrated their ability to enhance nutrient bioavailability [2], modulate gut microbial ecology [3], and augment gut barrier competence via short-chain fatty acid (SCFA)-mediated metabolic pathways [4,5]. These multifaceted effects underscore their potential as both therapeutic and productivity-enhancing agents [6]. Lactic acid bacteria, including *Lactobacillus*, *Bifidobacterium*, *Pediococcus*, *Leuconostoc*, etc., can produce lactic acid and bacteriocins, induce immune responses, prevent infections, and support intestinal health [7,8]. Multiple lactic acid bacteria have pH and bile salt tolerance and are used as feed additives [9].

The nursery period represents a pivotal developmental stage within swine production, characterized by significant physiological and metabolic challenges. This period is often associated with intestinal dysfunction, resulting in reduced health status, feed intake, and growth rates [10]. Probiotics play a pivotal role during this vulnerable stage by concurrently supporting nutrient absorption and maintaining intestinal barrier function. For instance, *Lactobacillus* and *Pediococcus* improve villus morphology and reduce diarrhea incidence [11,12], while *Bifidobacterium* alleviates intestinal dysfunction and decreases pathogen load [13]. The development and application of probiotic strategies have become integral to modern intensive swine production systems [14]. Traditional delivery methods, such as oral gavage or dry feed additives, face practical limitations, including inconsistent dosing, increased labor requirements, and elevated animal stress. Liquid feeding (LF) systems offer a promising alternative by facilitating cost-effective nutrient hydration and precise additive distribution [15]. Additionally, LF may enhance probiotic efficacy by improving microbial viability and prolonging mucosal contact. However, the mechanistic interplay between probiotics and LF systems remains insufficiently characterized.

Therefore, we developed a novel feeding regimen incorporating probiotics into LF. Using a multi-omics approach, we evaluated three dietary regimens: solid feed (Dry), liquid feed (Liq), and probiotic-enriched liquid feed (Pro), comparing their impacts on growth metrics, serum metabolic profiles, intestinal microbiome configuration, and metabolomic pathway dynamics in nursery piglets. Our findings elucidate the mechanistic basis of probiotic efficacy in LF systems and identify key microbial and metabolic biomarkers associated with growth optimization.

## 2. Results

### 2.1. Growth Performance and Nutrient Digestibility

Growth data across dietary regimens are presented in Table 1. While the Liq group showed slight improvements in growth and feed intake compared to the Dry group, these discrepancies failed to reach statistical significance (*p* > 0.05). In contrast, supplementing the liquid feed with *B. infantis* (1.11 × 10^5^ CFU/mL), *L. plantarum* (4.25 × 10^6^ CFU/mL), and *P. acidilactici* (2.50 × 10^6^ CFU/mL) significantly improved multiple growth performance indicators. Specifically, the Pro group exhibited a 20.00% (*p* < 0.01) and 17.86% (*p* < 0.01) increase in average daily gain (ADG), and an 8.91% (*p* < 0.01) and 6.08% (*p* < 0.05) increase in average daily feed intake (ADFI), relative to the Dry and Liq groups, respectively. Moreover, the feed conversion ratio (FCR) was reduced by 8.08% (*p* = 0.07) and 8.78% (*p* < 0.05), respectively. Table 2 demonstrates that probiotic supplementation partially enhanced the ATTD of select nutrients, with either extract showing a marginally significant improvement (*p* = 0.08).

### 2.2. Serum Biochemistry and Intestinal Permeability

Serum biochemical analysis indicated distinct variations among the groups (Figure 1A). Compared with the Dry group, significantly elevated levels of TP (*p* < 0.01) and IgG (*p* < 0.05) were observed in the Liq group. Additionally, NEFA concentrations in the Pro group were markedly higher relative to the other two groups (*p* < 0.05), with corresponding TG concentrations notably lower than those in the Liq group. Furthermore, probiotic supplementation resulted in a significant reduction in iFABP levels (*p* < 0.05), although the observed decreases in DAO, DLA, and LPS levels were not statistically significant (*p* > 0.05). These findings suggest that probiotic administration might contribute to a reduction in intestinal permeability and attenuating intestinal damage (Figure 1B).

### 2.3. Barrier Function and Colonic SCFA

Tight junction mRNA expression was analyzed in the jejunum and colon (Figure 2A). The Liq group showed a significantly enhanced jejunal expression of *ZO-1* (*p* < 0.05), as well as an increased expression of *Mucin2* in the jejunum and colon compared to the Dry group (*p* < 0.05). Furthermore, supplementation with probiotics markedly elevated the expression of *ZO-1*, *Claudin-1*, and *Occludin* specifically within the jejunum (*p* < 0.05), while no notable changes were detected in the colon. Colonic SCFA analysis showed that total SCFA concentration and levels of individual acids including acetic, propionic, butyric, isobutyric, and isovaleric acids were significantly elevated in the Pro group relative to both Liq and Dry groups (*p* < 0.05) (Figure 2B).

### 2.4. Gut Microbiota

Colonic content samples collected at trial initiation (Day 0) and termination (Day 60) underwent 16S rRNA gene sequencing. The sequencing analysis indicated a marked increase in microbial diversity associated with age (*p* < 0.05) (Figure 3A). PCoA analysis revealed minimal baseline variation (Day 0), with clear clustering emerging post-intervention (PERMANOVA R^2^ = 0.32, *p* < 0.01) (Figure 3B), corroborated by hierarchical clustering patterns (Appendix A). Age and feeding regime accounted for 18.35% (*p* < 0.01) and 6.69% (*p* = 0.06) of the variance in microbial composition, respectively (Appendix A). Major bacterial genera included *Lactobacillus*, *norank_f_Muribaculaceae*, *norank_f_Eubacterium_coprostanoligenes_group*, etc. (Figure 3C). Excluding the Pro group for the assessment of age–feed interaction, the Venn diagram showed 198 genera present at both time points, of which 25% disappeared over time (Figure 3D). Microbiome Multivariate Association with Linear Models (MaAsLin2) analysis was performed on 169 bacterial genera exhibiting an average relative abundance exceeding 0.01% across all samples, with age (Day 0/Day 60) and feed type (solid feeding/liquid feeding) included as independent variables. The analysis revealed that 92 bacterial genera exhibited significant responses to age, whereas 16 genera were associated with feed type (Figure 3E,F). Signature associations included *Rikenellaceae_RC9_gut_group*, *Pseudobutyrivibrio*, *Anaerorhabdus_furcosa_group* and *Lactobacillus* with Dry group, versus *Corynebacterium*, *Oscillospira*, *Romboutsia,* and *Ruminiclostridium* in Liq (Figure 3G).

To investigate the specific effects of the probiotics, we analyzed the microbial data from Day 60. The Sobs and Shannon indices of gut microbiota did not differ significantly among the piglet groups; however, the Simpson index in the Pro group was significantly higher than that in the Liq group (*p* < 0.05) (Figure 4A). PCoA analysis exhibited pronounced differences in microbial community composition among groups (PERMANOVA R^2^ = 0.22, *p* < 0.01) (Figure 4B). Probiotic supplementation and feed type accounted for 12.19% (*p* < 0.01) and 9.90% (*p* < 0.01) of the variance in microbial composition, respectively (Appendix A).

MaAsLin2 analysis, using probiotic supplementation (yes/no) and feed type (solid feeding/liquid feeding) as independent variables, revealed that Lactobacillus and *Bifido-bacterium* were significantly associated with probiotic supplementation, mitigating or slowing their age-related decline (Figure 4C,D). Absolute quantitative analysis further corroborated these findings (Figure 4E). In addition, certain bacterial genera, such as *Rikenellaceae_RC9_gut_group* and *Olsenella*, exhibited negative correlations with liquid feeding but were responsive to probiotic treatment. In contrast, genera such as *Corynebacterium* and *Ruminiclostridium* showed the opposite pattern (Figure 4F). Lefse analysis confirmed these findings and further showed that probiotics enriched genera such as *Eubacterium_ventrisun_group* and *Lactococcus*, while inhibiting the growth of *Streptococcus* and *Mogibacterium* (Appendix A). The correlation network suggested significant associations between *Lactobacillus*, *Bifidobacterium* and other key genera (Figure 4G).

Furthermore, elevated abundances of *Lactobacillus*, *Bifidobacterium*, and *Lactococcus* observed within the Pro group exhibited positive associations with growth performance, serum NEFA concentrations, and colonic SCFA levels. Conversely, microbial genera predominantly present in the Dry group demonstrated inverse correlations with serum TP, whereas genera enriched in the Liq group correlated positively with both serum TP and TG levels (Figure 4H). These differential correlations underscore distinct metabolic profiles elicited by each dietary intervention.

### 2.5. Gut Metabolism

Metabolomic profiling was used to investigate how different feeding regimens affected the intestinal metabolome of nursery piglets. PCA and partial least squares discriminant analysis (PLS-DA) revealed distinct differences in metabolite profiles between the groups (Figure 5A,B). Further analysis using OPLS-DA highlighted the significant impact of feed type and probiotic supplementation on intestinal metabolism (Figure 5C). A volcano plot analysis identified several differential metabolites between the Dry and Liq groups, as well as between the Liq and Pro groups (Figure 5D).

KEGG pathway enrichment demonstrated LF downregulation in amino acid/lipid metabolism pathways (cutin, suberine and wax biosynthesis, and linoleic acid metabolism), particularly tryptophan metabolism (Liq vs. Dry: *p* < 0.05, DA score = -0.15). Probiotic supplementation reversed this trend, upregulating tryptophan metabolism (Pro vs. Liq: *p* < 0.01, DA score = 0.46) (Figure 5E,F). An analysis of specific metabolites within the tryptophan pathway confirmed an increase in xanthurenic acid and kynurenic acid, with a concomitant decrease in 3-methylindole (skatole) in the Pro group (Figure 5G).

Procrustes analysis showed a strong alignment between the trends in differential metabolite levels, phenotypic traits, and microbial abundance across the feeding groups (Figure 6A). Correlation analysis indicated that xanthurenic acid, kynurenic acid, and formyl anthranilic acid were positively associated with improved growth performance and higher colonic SCFA levels, while 3-methylindole showed a negative correlation (Figure 6B). ROC analysis further demonstrated that these metabolites may serve as reliable biomarkers for evaluating the metabolic effects of various feeding strategies, with area under the curve (AUC) values exceeding 0.9 (Figure 6C). The O2PLS-DA model revealed a clear separation between the Liq and Pro groups, with relatively compact groupings for both microbial and metabolite samples, suggesting that probiotic supplementation affected both microbial and metabolic profiles in a consistent manner. High loading scores of *Lactobacillus* and *Bifidobacterium* indicated their key role in distinguishing the groups (Figure 6D,E). The two-factor network diagram confirmed the positive correlations between *Bifidobacterium* and *Lactobacillus* with kynurenic acid and xanthurenic acid, alongside *Lactobacillus*-mediated skatole (Figure 6F). These observations highlight the significant contributions of *Bifidobacterium* and *Lactobacillus* to the regulation of intestinal metabolic activity in nursery piglets.

### 2.6. Regulatory Effects of Probiotics on Tryptophan Metabolism

To further explore the direct regulatory effects of probiotics on tryptophan metabolism, serum samples from the Liq and Pro groups were subjected to targeted metabolomic analysis. PCA and OPLS-DA revealed a distinct separation in the serum tryptophan metabolic profiles between the Pro and Liq groups (Figure 7A). Analysis of the top 15 metabolites based on VIP scores indicated that the levels of xanthurenic acid (*p* < 0.05) and 3-hydroxykynurenine (*p* > 0.05) were higher in the Pro group compared to the Liq group, consistent with previous findings and confirming the activation of the kynurenine pathway by probiotics (Figure 7B). Further analysis demonstrated that the serum levels of L-tryptophan (L-TRP), indole-3-lactic acid (ILA), and related metabolites were significantly elevated in the Pro group piglets (*p* < 0.01). The observed decrease in colonic L-TRP levels and increase in ILA concentrations in the Pro group indicated that probiotic supplementation promoted tryptophan metabolism and absorption (*p* < 0.01) (Figure 7C,D). Spearman’s correlation analysis revealed a significant positive association between *Lactobacillus* and serum levels of L-TRP (R = 0.83, *p* < 0.01) and ILA (R = 0.80, *p* < 0.01). A significant positive correlation was also observed between *Bifidobacterium* and serum ILA concentrations (R = 0.70, *p* < 0.05) (Figure 7E). To verify the potential contribution of probiotics to tryptophan metabolism, *B. infantis* (BI), *L. plantarum* (LP), and their coculture (BI + LP) were incubated in an inorganic salt medium containing L-TRP as the sole carbon source. The results demonstrated that all groups efficiently metabolized L-TRP to produce ILA, with ILA production following the metabolic efficiency trend of L-TRP utilization, namely BI + LP > LP > BI (*p* < 0.01).

## 3. Discussion

Probiotic-enriched liquid feeding effectively improved the growth performance of nursery piglets, with ADG increasing by over 17.86% and FCR reducing by up to 8.08%. These improvements were accompanied by higher colonic SCFA levels and enhanced expression of tight junction proteins. In addition, colonization by *B. infantis* and *L. plantarum* reshaped gut microbiota and promoted tryptophan metabolism and ILA production. This integrated feeding strategy outperforms conventional liquid or solid feed approaches, supporting the feasibility of probiotic delivery through LF systems.

LF offers advantages over solid feeding during the nursery period, such as improved feed intake and efficiency by facilitating the transition from milk to feed [16]. However, our results showed that LF alone did not improve growth performance or nutrient digestibility. This may be due to the piglets’ immature gut adaptation to feed-to-water ratios and the inherent limitations of LF systems, including nutrient sedimentation and particle heterogeneity [17]. In practice, complete pelleted feed is often mixed directly with water rather than using LF-specific formulations, which may compromise nutrient uniformity and reduce production efficiency. Similarly, a study reported no improvement in performance with liquid feeding of corn-based diets, likely due to sedimentation, low water-binding capacity, and lack of endogenous phytase [18]. Additionally, fermentation is inevitable in LF systems [19], potentially increasing undesirable microbes, compromising feed quality and palatability, and leading to nutrient loss [20].

In contrast, the Pro group exhibited significant improvements in ADG, ADFI, and feed efficiency, along with a tendency toward improved digestibility of several nutrients. Previous reports have shown that probiotic supplementation can increase ADG by 8–12% and reduce FCR by 5–7%, resulting in economic savings of approximately USD 5–10 per pig [21]. The inclusion of lactic acid bacteria also helped prevent undesirable fermentation in LF systems, thereby reducing feed spoilage and maintenance costs. Moreover, probiotic supplementation markedly enhanced intestinal barrier function. These findings align with prior studies indicating probiotics’ effectiveness in promoting gut integrity and facilitating nutrient absorption [22,23,24]. Elevated SCFA levels in the Pro group further support these observations, as SCFA play a key role in maintaining mucosal integrity and supplying energy to colonic cells [25,26].

Microbial composition was primarily influenced by age, accounting for more than half of the changes in bacterial genera. However, probiotic supplementation and feed type also significantly influenced gut microbial structure, explaining 12.19% (*p* < 0.01) and 9.90% (*p* < 0.01) of the observed differences, respectively. As reported in previous studies, individual-level factors, host genotypes, and environmental conditions collectively contribute to gut microbiota variability and may account for the remaining unexplained variation [27,28]. The Liq group showed a concomitant enrichment of proteolytic taxa such as *Terrisporobacter* and *Alistipes* [29,30], alongside the proliferation of opportunistic pathogens including *Streptococcus* and *Corynebacterium*, both associated with systemic infections [31,32]. Probiotic supplementation effectively suppressed these pathogens while promoting beneficial genera such as *Lactobacillus* and *Bifidobacterium*. Colonization with these probiotics further enriched genera like *Lactococcus*, *Rikenellaceae_RC9_gut group*, and *Eubacterium ventriosum group*, which are known to enhance SCFA production and exert anti-inflammatory effects [33,34,35]. Notably, the absence of *Pediococcus* colonization underscores strain-specific limitations in viability and ecological competitiveness under liquid feeding conditions. Although *P. acidilactici* was included for its recognized probiotic properties, it is not a dominant member of the porcine gut microbiota [36]. Moreover, the transition to an anaerobic intestinal environment during the nursery phase may have further hindered its colonization [37]. The probiotic efficacy of *Pediococcus* appears to be strain-dependent: while *P. pentosaceus* CACC616 contributes to gut health, others, such as *P. acidilactici*, exhibit limited persistence under ETEC challenge [38,39].

Probiotic supplementation significantly influenced colonic metabolism, with notable associations observed in tryptophan metabolism, a pathway crucial for gut health, immune regulation, and the gut–brain axis [40,41,42]. Pathway enrichment analysis showed that tryptophan metabolism was significantly suppressed in the Liq group compared to the Dry group but was substantially restored with probiotic supplementation. This restoration coincided with increased levels of xanthurenic acid, kynurenic acid, and related intermediates in the Pro group, suggesting that probiotics may modulate the kynurenine pathway. These findings align with previous research demonstrating that gut microbiota modulate tryptophan availability and the activity of the downstream kynurenine pathway, thereby influencing both neurological and gastrointestinal functions [43,44,45]. Probiotics, including *Lactobacillus* spp. and *Bifidobacterium infantis*, have demonstrated beneficial effects in alleviating intestinal inflammation and reinforcing barrier integrity [46,47]. These beneficial impacts have been linked to decreased kynurenine levels coupled with elevated kynurenic acid concentrations [48,49]. However, it remains unclear whether these effects are primarily driven by microbial enzymatic activity or host-regulated mechanisms such as indoleamine 2,3-dioxygenase (IDO1) expression [50,51]. Additionally, probiotics may lower 3-methylindole (skatole), a toxic metabolite associated with intestinal apoptosis and inflammation, further supporting gut health [52,53]. The positive correlations observed between tryptophan metabolites, beneficial bacterial genera, and growth performance emphasize the role of *Lactobacillus* and *Bifidobacterium* in driving metabolic changes that enhance growth and intestinal health.

Furthermore, serum-targeted metabolomics revealed that probiotic supplementation significantly altered the circulating profile of tryptophan metabolism in piglets. The observed reduction in colonic tryptophan levels, along with increased serum concentrations, further confirms the regulatory role of probiotic supplementation in modulating tryptophan availability and metabolism [54]. These findings are consistent with previous studies showing that supplementation with *Bifidobacterium* and *Lactobacillus reuteri* enhances the production of downstream metabolites through the tryptophan catabolic pathway [55,56]. Importantly, both in vitro and in vivo studies have confirmed that *B. infantis* and *L. plantarum* efficiently metabolize tryptophan to generate ILA, leading to its accumulation in piglet serum. As early as 1979, Aroagozzini et al. [57] reported that 51 strains of *Bifidobacterium* could convert tryptophan to ILA under resting conditions, a process dependent on the aromatic lactate dehydrogenase (Aldh) gene cluster, including *AldH1* and *AldH2* [58]. *L. plantarum* may produce the intermediate indole-3-pyruvic acid (IPYA) via the tryptophan transaminase pathway, which is subsequently converted to ILA through a decarboxylation reaction [59]. ILA is a key metabolite of probiotic-derived tryptophan metabolism that activates aryl hydrocarbon receptor (AhR) and promotes anti-inflammatory and gut-repair functions [60,61]. These effects may contribute to the health-promoting roles of *B. infantis* and *L. plantarum*, although the underlying mechanisms remain to be elucidated.

## 4. Materials and Methods

### 4.1. Animals, Diets, and Experimental Design

A total of 270 piglets (Duroc × Landrace × Yorkshire) were transferred to a nursery equipped with liquid feeding tubes and troughs after reaching an appropriate weight of 16.84 ± 0.12 kg at 45.89 ± 3.80 days of age. The piglets were randomized into three dietary treatment groups: solid feed group (Dry), liquid feed group (Liq), or probiotic-enriched liquid feed group (Pro). Each dietary regimen included 6 pens, with 15 piglets housed per pen.

The experiment took place on a commercial liquid-fed pig farm in Anhui, China. The formal feeding trial began in March and lasted for 60 days. All experimental pens (4.5 m × 3.5 m) were equipped with polyvinyl chloride (PVC) slatted flooring and automated stainless-steel nipple watering drinkers. Throughout the trial, environmental conditions were maintained at 22–24 °C and 60–70% humidity.

All dietary treatment groups received the same pelleted basal diet designed in compliance with NRC (2012) nutrient specifications. The diet was administered three times daily at 09:00, 16:00, and 21:00. Complete formulation details including ingredient and nutrient determination are presented in Table 3.

The LF was prepared by homogenizing the basal diet with water at a 1:3.5 (*w*/*v*) ratio. The probiotic-enriched liquid feed was prepared by supplementing a compound probiotic powder immediately before the 09:00 feeding. A total of 500 g of the compound probiotic powder was added per ton of LF, containing *Bifidobacterium infantis* (5.57 × 10^8^ CFU/g), *Lactobacillus plantarum* (4.25 × 10^10^ CFU/g), and *Pediococcus acidilactici* (1.25 × 10^10^ CFU/g), at a ratio of 2:1:1 (*w*/*w*/*w*). Accordingly, the theoretical concentrations of *B. infantis*, *L. plantarum*, and *P. acidilactici* the LF were 1.11 × 10^5^ CFU/mL, 4.25 × 10^6^ CFU/mL, and 2.50 × 10^6^ CFU/mL, respectively. Two types of LF were provided through an automatic system (DeBa Brothers Machinery Co., Ltd., Qingdao, Shandong, China) with feed intake monitored by infrared sensors. The Dry group was fed manually in the same pens with the feed portals closed. Feed was weighed daily to estimate intake and wastage.

### 4.2. Sample Collection

Basal diet samples from each growth stage were collected for chemical analysis. Intestinal samples were collected following pen sanitation at 10:00. At the beginning of the experiment, colon contents were collected from piglets via rectal swabs for microbiological analysis. During the final three days of the experiment, fresh feces were collected immediately after defecation for the determination of apparent total tract digestibility (ATTD). To stabilize nitrogen, 10 mL of a 10% sulfuric acid (H_2_SO_4_) solution was introduced per 100 g of fecal material. All samples were subsequently stored at −20 °C until further analysis.

After overnight fasting upon completion of the trial, one piglet from each pen was randomly chosen for slaughter. Blood samples were allowed to clot at room temperature for 30 min and subsequently centrifuged (3000 rpm, 4 °C) for 10 min to obtain serum. After the intestine was removed, each intestinal segment was quickly ligated and contents from the midsection of the colon were collected. Tissue samples from the jejunum and colon were excised 20 cm from the anterior end, rinsed with PBS, and immediately snap-frozen in liquid nitrogen for subsequent analyses.

### 4.3. Nutrient Determination and Apparent Total Tract Digestibility

For the calculation of apparent total tract digestibility (ATTD), fecal samples collected during the final three days of the experimental period were pooled. Subsequently, basal diet and fecal samples underwent drying at 65 °C for 72 h, followed by milling using a coffee grinder and passing through a 1 mm sieve.

The nutrient composition was assessed in accordance with protocols from the Association of Official Analytical Chemists (AOAC) [62]: dry matter (method 930.15), ether extract (method 920.39), and crude protein (method 990.03). Additionally, total amino acids [63], calcium [64], and total phosphorus [65] concentrations were quantified following the Chinese National Standard. Gross energy was assessed utilizing an adiabatic oxygen bomb calorimeter (Parr Instrument Co., Moline, IL, USA).

Acid insoluble ash (AIA) served as the endogenous marker for determining ATTD, according to the Chinese National Standard [66]. The calculation for ATTD (%) was conducted as follows:ATTD (%) = 100 × [1 − (marker D/marker F) × (nutrient F/nutrient D)],
where marker D and marker F represent the concentrations of AIA in the diet and feces, respectively, and nutrients D and F denote the nutrient levels in the diet and feces.

### 4.4. Analysis of Serum Parameters

Serum samples were evaluated for total protein (TP), albumin (ALB), glucose (GLU), total cholesterol (T-CHO), triglycerides (TG), high-density lipoprotein (HDL-C), low-density lipoprotein (LDL-C), and non-esterified fatty acids (NEFA) using an automated serum biochemistry analyzer. Immunoglobulins (IgG, IgM, IgA), diamine oxidase (DAO), D-lactate (DLA), lipopolysaccharides (LPS), and intestinal fatty acid binding protein (iFABP) were quantified using ELISA kits (YOBIBIO, Shanghai, China).

### 4.5. Real-Time Quantitative PCR

Jejunal and colonic total RNA was isolated using Trizol reagent (Takara), and the concentrations of extracted RNA were quantified via NanoDrop spectrophotometry (Thermo Scientific, Waltham, MA, USA). Complementary DNA (cDNA) synthesis was performed with random primers utilizing an RT kit (Takara, San Jose, CA, USA). Real-time quantitative PCR was subsequently conducted employing SYBR Green Master Mix (Takara), and the relative expression of target genes was evaluated by the 2^−ΔΔCT^ method. Detailed sequences of primers employed are listed in Table 4.

### 4.6. Colonic SCFA Analysis

Gas chromatography was used for the quantification of short-chain fatty acids (SCFA). Briefly, fecal samples (0.1–0.2 g) were combined with 600 µL methanol, adjusted to pH 2–3, vortex-mixed, and then, centrifuged. The resultant supernatant underwent filtration and analysis via a Varian CP-3800 gas chromatograph (Richmond, VA, USA).

### 4.7. 16S rRNA Sequencing

Genomic DNA was isolated from colonic content samples (*n* = 6 per group), employing the FastPure Stool DNA Isolation Kit (MJYH, Shanghai, China). PCR amplification targeting the bacterial 16S rRNA gene was performed using primers 338F (5′-ACTCCTACGGGAGGCAGCAG-3′) and 806R (5′-GGACTACHVGGGTWTCTAAT-3′), corresponding to the V3-V4 region. Sequencing of the amplified products was conducted on an Illumina Nextseq2000 platform (Illumina, San Diego, CA, USA) by Majorbio Bio-Pharm Technology Co., Ltd. (Shanghai, China). Sequence data underwent analysis using the QIIME2 pipeline, involving denoising through DADA2 and chimera elimination via VSEARCH. Resulting amplicon sequence variants (ASVs) were generated from high-quality sequences and clustered based on tags, excluding non-biological nucleotide sequences.

Bioinformatics processing was carried out using the Majorbio Cloud (https://cloud.majorbio.com, accessed on 15 June 2024). Alpha diversity was assessed via Chao1, Shannon, and Simpson indices based on the ASVs. Principal coordinate analysis (PCoA) and hierarchical clustering using Bray–Curtis distance were applied to compare microbial community composition between samples. The PERMANOVA test evaluated the variation explained by treatment groups and its statistical significance. Genera associated with age, feed type, and probiotic supplementation were identified using Microbiome Multivariate Association with Linear Models 2 (MaAsLin2) (https://huttenhower.sph.harvard.edu/tools, accessed on 9 July 2024).

### 4.8. Untargeted Metabolomic Analysis

Colonic contents were homogenized and extracted using a methanol–water mixture (4:1, *v*/*v*). For LC-MS/MS analysis, supernatants were transferred to Majorbio Bio-Pharm Technology Co., Ltd. (Shanghai, China). Chromatographic separation utilized a Thermo Scientific UltiMate 3000 HPLC system, while metabolites were detected using a Q-Exactive high-resolution tandem mass spectrometer (Thermo Scientific) equipped with an ACQUITY UPLC BEH C18 column (Waters, London, UK). Raw data were transformed into the mzXML format via ProteoWizard (msConvert, version 3.0), and then analyzed using XCMS software (version 3.4)for peak detection and grouping. Metabolites were annotated using CAMERA and identified with MetaX software (http://metax.genomics.cn, accessed on 17 July 2024) by referencing both primary and secondary mass spectrometry data against a customized database.

The statistical evaluation utilized the aforementioned Majorbio Cloud. Metabolite profiles were assessed through principal component analysis (PCA) and orthogonal partial least squares discriminant analysis (OPLS-DA). According to the OPLS-DA model, identify significant metabolites whose Variable Importance in Projection (VIP) > 2 and statistical significance (*p* < 0.05). Pathway enrichment and analysis of differential metabolites were conducted using the KEGG database. Procrustes analysis was conducted to explore correlations between differential microbiota, metabolites, and phenotypic traits. Potential biomarkers were identified and validated through receiver operating characteristic (ROC) curves and O2PLS-DA model loading scores.

### 4.9. Targeted Metabolomic Analysis

Targeted metabolomic analysis of 38 tryptophan-related metabolites was performed using ultra-high-performance liquid chromatography coupled with tandem mass spectrometry (UHPLC–MS/MS). Serum samples collected from the Liq and Pro groups were thawed on ice, mixed with methanol containing internal standards (10 μL, 250 ng/mL), vortexed, incubated at −20 °C for protein precipitation, and centrifuged at 12,000 rpm for 10 min at 4 °C. Supernatants were transferred to autosampler vials for analysis.

Chromatographic separation was conducted on an ACQUITY UPLC HSS T3 column (1.8 μm, 100 mm × 2.1 mm i.d., Waters) using an ExionLC™ AD system (SCIEX). The mobile phases consisted of 0.1% formic acid in water (A) and 0.1% formic acid in acetonitrile (B), with a flow rate of 0.35 mL/min and a gradient from 90:10 to 5:95 (A:B) over 7 min. The injection volume was 2 μL and the column temperature was maintained at 40 °C. Mass spectrometric detection was carried out on a QTRAP 6500 + (SCIEX) equipped with an electrospray ionization (ESI) source operating in both positive and negative ion modes (+5500 V/−4500 V), with a source temperature of 550 °C and curtain gas pressure at 35 psi. Data acquisition was performed in multiple reaction monitoring (MRM) mode using optimized parameters for each metabolite. Raw data were processed using Analyst 1.6.3 and MultiQuant 3.0.3 software. Metabolites were identified and quantified based on retention time and peak shape compared with authentic standards. Data were log-transformed and further analyzed using MetaboAnalyst v5.0 for statistical evaluation and visualization.

### 4.10. In Vitro Metabolism of Tryptophan to Indole-3-Lactic Acid by Probiotics

Selected probiotic strains (*L. plantarum* and *B. infantis*) were inoculated into a liquid culture medium and incubated anaerobically at 37 °C for 12–24 h until reaching the stationary phase. Bacterial cultures were harvested by centrifugation at 8000 rpm for 5 min at 4 °C, and the pellets were washed twice with sterile phosphate-buffered saline (PBS, pH 7.4). The washed cells were resuspended in 1 mL of inorganic salt medium supplemented with 1 mM L-tryptophan (L-Trp), with three biological replicates per strain. The cultures were incubated anaerobically at 37 °C for 72 h. After incubation, the bacterial suspensions were centrifuged at 8000 rpm for 10 min at 4 °C, and the resulting supernatants were collected and stored at −80 °C for subsequent analysis. The composition of the inorganic salt medium is listed in Appendix A.

For metabolite quantification, 100 μL of each fermentation supernatant was mixed with 800 μL of prechilled methanol to precipitate proteins. The mixture was vortexed for 10 s and incubated on ice for 30 min or at −20 °C for 10 min. After centrifugation at 15,000 rpm for 10 min at 4 °C, the supernatant was collected and filtered through a 0.45 μm organic-compatible membrane. Concentrations of L-Trp and indole-3-lactic acid (ILA) were quantified using the subsequent chromatographic and mass spectrometry protocols.

### 4.11. Determination of Tryptophan and Indole-3-Lactic Acid in Colonic Samples

Approximately 50–100 mg of frozen colonic content was homogenized in 900 μL of 100% methanol and processed using the same protocol as for fermentation supernatants. Standard solutions were prepared in 80% methanol.

L-Trp concentrations were determined by high-performance liquid chromatography with ultraviolet detection (HPLC-UV). Separation was achieved using a Welch Ultimate C18 column (4.6 × 250 mm, 3 μm) with a mobile phase consisting of sodium acetate buffer and acetonitrile (85:15, *v*/*v*). The flow rate was 1.0 mL/min, the column temperature was maintained at 40 °C, and detection was performed at 282 nm. The injection volume was 10 μL.

ILA levels were quantified by ultra-performance liquid chromatography coupled with multiple reaction monitoring (UPLC-MRM). Chromatographic separation was performed on an Acquity BEH C18 column (2.1 × 100 mm, 1.7 μm) with mobile phase A (0.1% formic acid in water) and mobile phase B (0.1% formic acid in methanol). The gradient elution program was as follows: 0–0.5 min, 90% A; 0.5–2.0 min, linear decrease to 5% A; 2.0–3.0 min, 5% A; 3.0–3.5 min, return to 90% A; 3.5–5.0 min, 90% A. The flow rate was 0.4 mL/min, with an injection volume of 8 μL, and the column temperature set at 40 °C.

Mass spectrometry was conducted in positive-ion mode using electrospray ionization. Nitrogen served as both the nebulizing and drying gas, and argon was used as the collision gas. ILA was detected via the ion transitions *m*/*z* 206.1 → 118.0 (quantifier ion, CE 24 eV) and *m*/*z* 206.1 → 160.1 (qualifier ion, CE 12 eV), with a cone voltage of 22 V.

### 4.12. Statistical Analysis

Data analysis was executed using SPSS 26.0 (IBM Corp., Armonk, NY, USA). Results are displayed as means with corresponding standard deviation (SD) or standard error (SEM). Differences among groups were statistically evaluated by applying unpaired *t*-tests, one-way ANOVA with False Discovery Rate (FDR) correction, or Wilcoxon rank-sum tests, with significance set at *p* < 0.05.

Correlations among microbiota, metabolites, and phenotypic characteristics were assessed using Spearman’s correlation tests in R (version 4.3.1). Cytoscape 3.10.2 was utilized for network visualization, incorporating only correlations with absolute coefficient values greater than 0.6 and significance levels of *p* < 0.05.

## 5. Conclusions

This study shows that incorporating probiotics into a liquid feeding system can improve nursery piglet growth and intestinal health through alterations in gut microbiota composition and tryptophan metabolic pathways. Specifically, liquid-fed probiotic administration upregulated tight junction protein expression, thereby reinforcing intestinal barrier function, and facilitated the colonization of *B. infantis* and *L. plantarum*, alongside reshaping microbial communities and elevating SCFA levels. Additionally, metabolomic analysis indicated that tryptophan metabolism was altered, notably through the activation of the kynurenine pathway and elevated levels of ILA in serum. In vitro assays verified that both *B. infantis* and *L. plantarum* are capable of converting L-TRP into ILA. Overall, these findings highlight the multifaceted advantages of probiotic supplementation, offering theoretical insights for refining probiotic usage strategies and advancing liquid feeding approaches, as well as supporting the design of probiotic-driven intestinal health programs to enhance sustainable swine production.

## Figures and Tables

**Figure 1 ijms-26-05698-f001:**
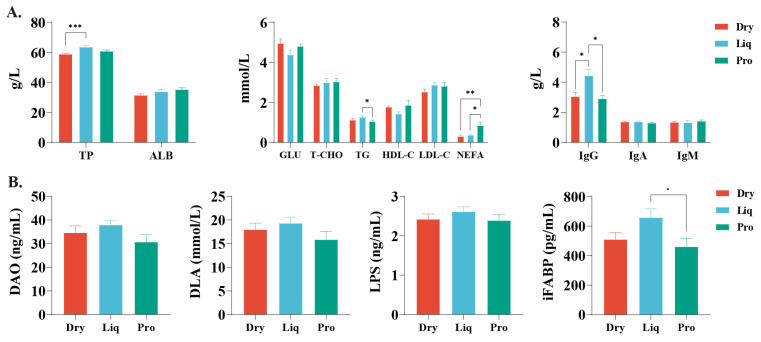
Effects of different feeding regimens on serum biochemistry and intestinal permeability. (**A**) Serum biochemistry. (**B**) Serum markers of intestinal permeability, including diamine oxidase (DAO), D-lactate (DLA), lipopolysaccharide (LPS), and intestinal fatty acid-binding protein (iFABP). Data expressed as mean (SEM). Dry = solid feed group; Liq = liquid feed group; Pro = probiotic-enriched liquid feed group. Asterisks indicate significant differences: * *p* < 0.05, ** *p* < 0.01, *** *p* < 0.001 (*n* = 6).

**Figure 2 ijms-26-05698-f002:**
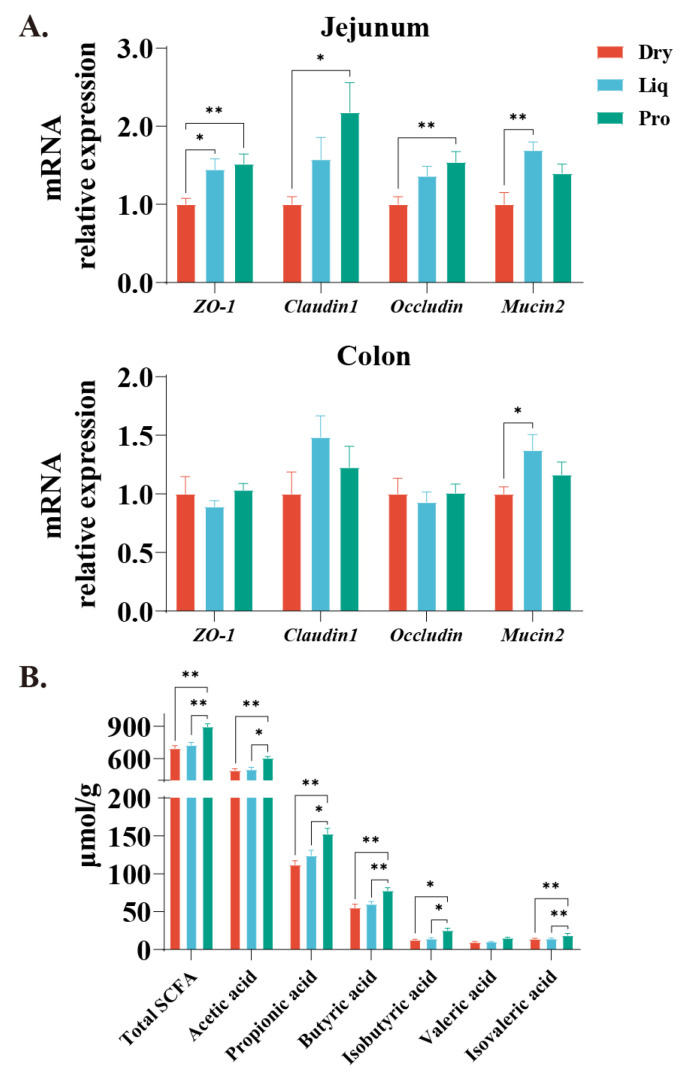
Effects of different feeding regimens on intestinal barrier function and colonic SCFA. (**A**) mRNA expression of tight junction proteins and mucoprotein in the jejunum and colon (*ZO-1*, *Claudin-1*, *Occludin* and *Mucin2*). (**B**) Colonic SCFA content. Data expressed as mean (SEM). Dry = solid feed group; Liq = liquid feed group; Pro = probiotic-enriched liquid feed group. Asterisks indicate significant differences: * *p* < 0.05, ** *p* < 0.01 (*n* = 6).

**Figure 3 ijms-26-05698-f003:**
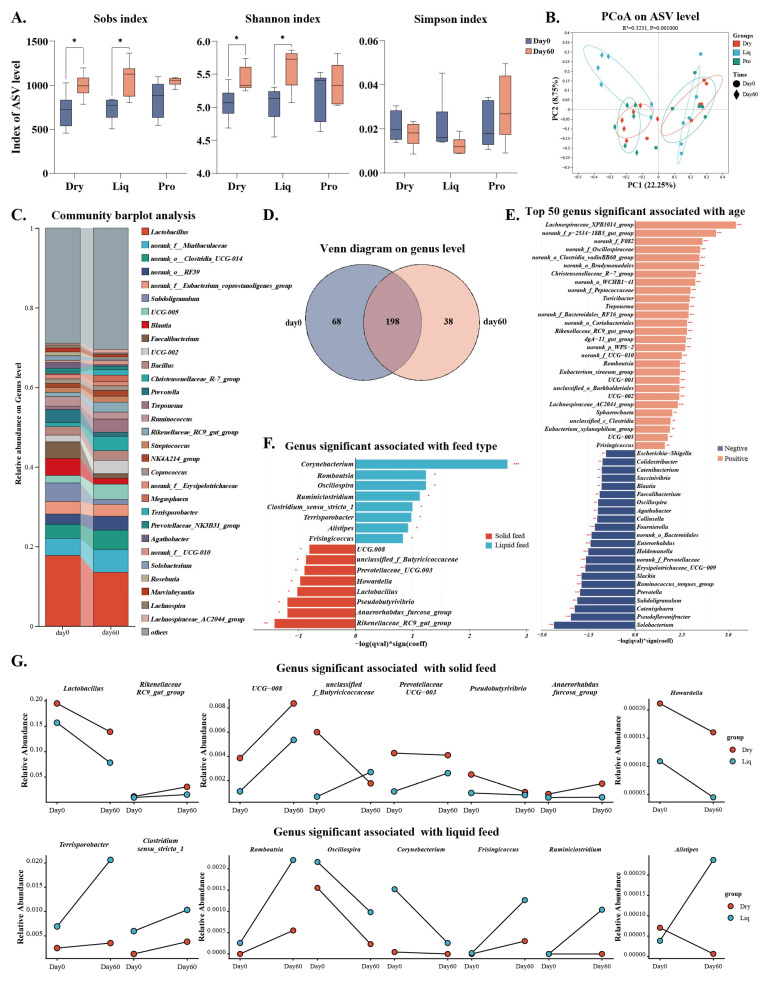
Effects of feed type on gut microbiota composition over time. (**A**) Changes in α-diversity over time. (**B**) PCoA plot of bacterial community composition based on Bray–Curtis distance. (**C**) Bar plots of bacterial genera at Day 0 and Day 60. (**D**) Venn diagram of genus present at Day 0 and Day 60. (**E**) Top 50 genus significantly associated with age (MaAsLin2). (**F**) Genus associations with feed type (MaAsLin2). (**G**) Changes in the abundance of specific genus enriched by solid or liquid feed. Dry = solid feed group; Liq = liquid feed group; Pro = probiotic-enriched liquid feed group; Day 0 = beginning of experiment; Day 60 = end of experiment. Asterisks indicate significant differences: * *p* < 0.05, ** *p* < 0.01, *** *p* < 0.001 (*n* = 6).

**Figure 4 ijms-26-05698-f004:**
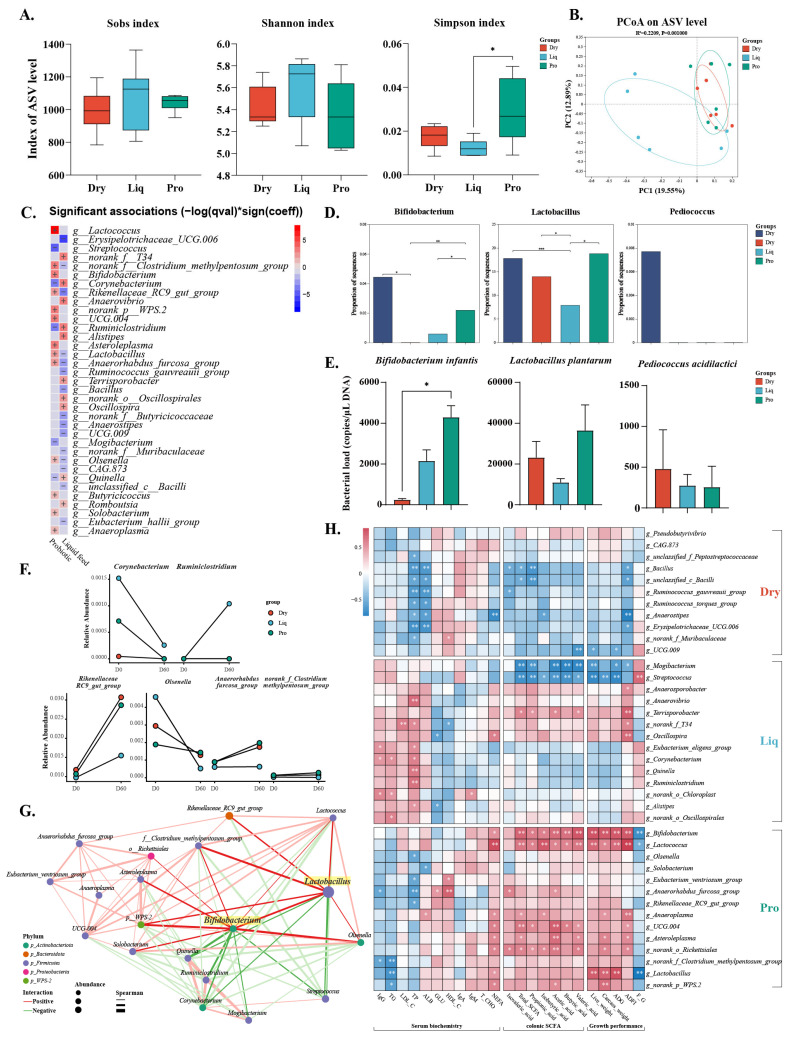
Effects of probiotic supplementation on gut microbiota. (**A**) Changes in α-diversity across groups. (**B**) PCoA plot of bacterial community composition based on Bray–Curtis distance. (**C**) Genus associations with feed type or probiotic supplementation (MaAsLin2). (**D**) Relative abundance of *Bifidobacterium*, *Lactobacillus*, and *Pediococcus*. (**E**) Absolute quantification of strains. (**F**) Changes in the abundance of specific genus enriched by liquid feed or probiotic supplementation. (**G**) Network analysis of differential genus using Spearman’s correlation (|r| > 0.6, *p* < 0.05). (**H**) Spearman’s correlation between differential genus and serum biochemistry, colonic SCFA, and growth performance. Dry = solid feed group; Liq = liquid feed group; Pro = probiotic-enriched liquid feed group; Day 0 = start of experiment; Day 60 = end of experiment. Asterisks indicate significant differences: * *p* < 0.05, ** *p* < 0.01, *** *p* < 0.001 (*n* = 6).

**Figure 5 ijms-26-05698-f005:**
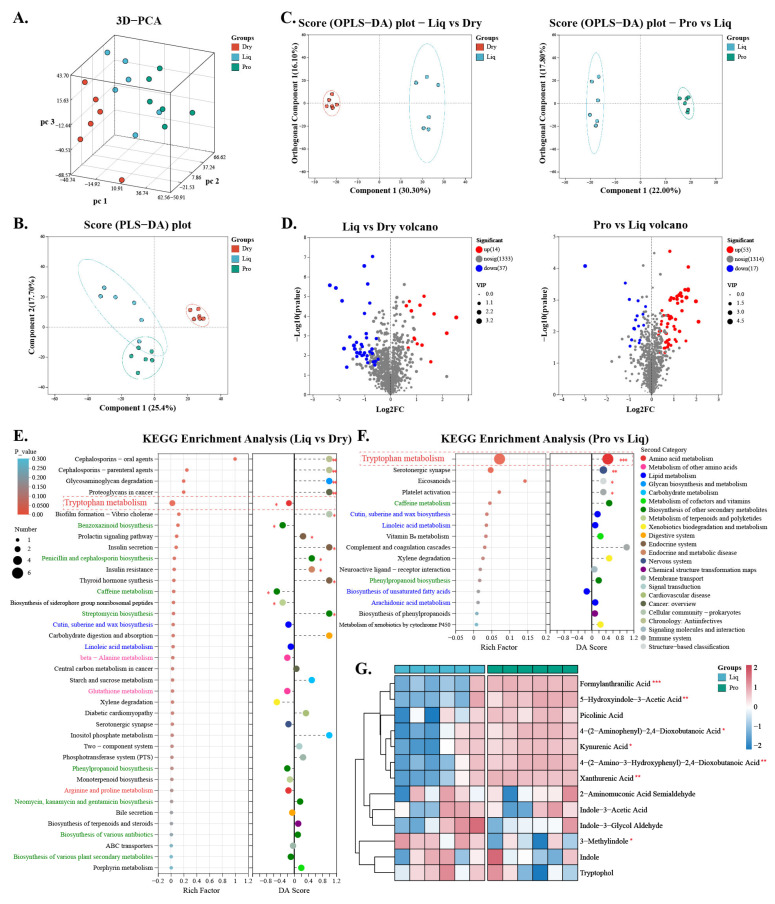
Metabolomic profiling and KEGG pathway analysis across groups. (**A**,**B**) PCA and PLS-DA analysis of intestinal metabolites identified by LC-MS/MS. (**C**) OPLS-DA of intestinal metabolites between Liq and Dry or Pro groups. (**D**) Volcano plot identifying differential metabolites (VIP > 2, *p* < 0.05). (**E**,**F**) KEGG pathway analysis of differential metabolites between (**E**) Liq vs. Dry and (**F**) Pro vs. Liq. (**G**) Heatmap of tryptophan-related metabolites in Liq and Pro groups. Dry = solid feed group; Liq = liquid feed group; Pro = probiotic-enriched liquid feed group. Asterisks indicate significant differences: * *p* < 0.05, ** *p* < 0.01, *** *p* < 0.001 (*n* = 6).

**Figure 6 ijms-26-05698-f006:**
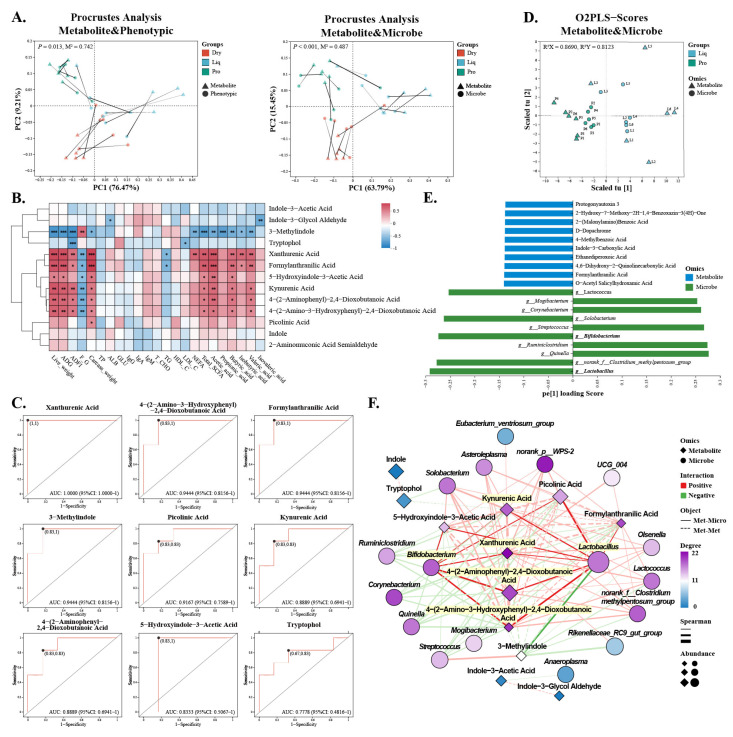
Correlations between metabolites, microbiota, and phenotypic traits. (**A**) Procrustes analysis of metabolites and microorganisms or phenotypes. (**B**) Spearman’s correlation between tryptophan-related metabolites and serum biochemistry, colonic SCFA, and growth performance. (**C**) ROC curves of tryptophan-related metabolites between Liq and Pro groups. (**D**,**E**) PCA plot and loading scores of O2PLS-DA model of metabolites and microbiota between Liq and Pro groups. (**F**) Network analysis of metabolites and microbiota by Spearman’s correlation (|r| > 0.6, *p* < 0.05). Dry = solid feed group; Liq = liquid feed group; Pro = probiotic-enriched liquid feed group; Day 0 = beginning of experiment; Day 60 = end of experiment. Asterisks indicate significant differences: * *p* < 0.05, ** *p* < 0.01, *** *p* < 0.001 (*n* = 6).

**Figure 7 ijms-26-05698-f007:**
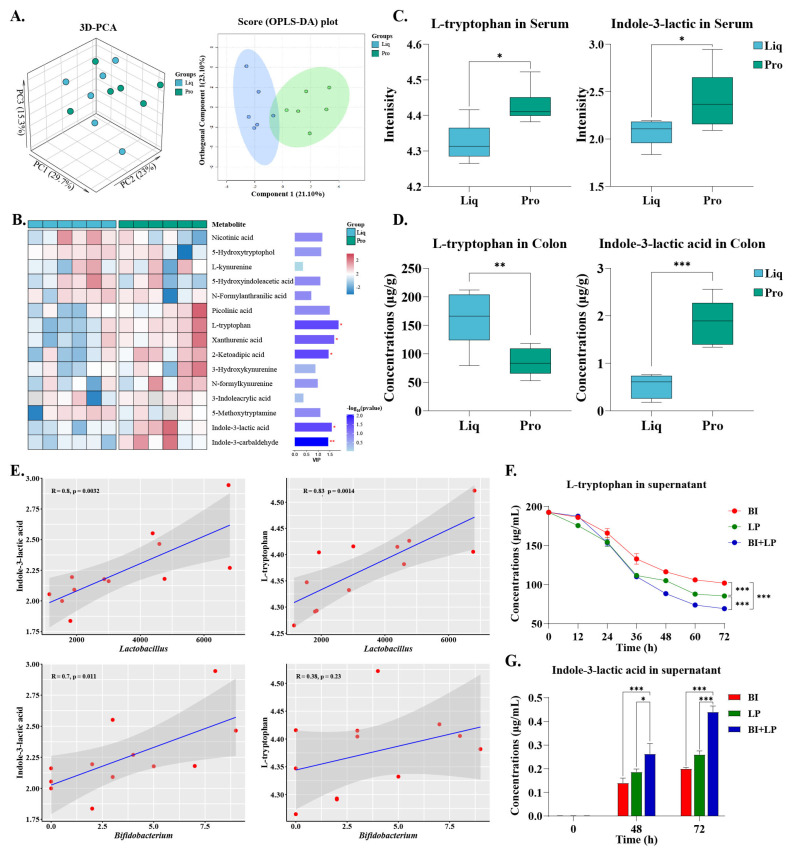
Regulatory effects of probiotics on tryptophan metabolism. (**A**) PCA and OPLS-DA analysis of serum tryptophan metabolism between Liq and Pro groups. (**B**) Top 15 differential metabolites ranked by VIP scores. (**C**) Signal intensities of L-TRP and ILA in piglet serum. (**D**) Concentrations of L-TRP and ILA in piglet colon. (**E**) Correlation between *Lactobacillus/Bifidobacterium* abundance and serum L-TRP/ILA levels. (**F**,**G**) In vitro L-TRP metabolism and ILA production by probiotic strains. Liq = liquid feed group; Pro = probiotic-enriched liquid feed group; BI = *Bifidobacterium infantis*; LP = *Lactobacillus plantarum*; BI + LP = coculture of *B. infantis* and *L. plantarum*. Asterisks indicate significant differences: * *p* < 0.05, ** *p* < 0.01, *** *p* < 0.001 (*n* = 6).

**Table 1 ijms-26-05698-t001:** Effects of different feeding regimes on growth performance in nursery piglets.

Item	Group			SEM	*p*-Value
	Dry	Liq	Pro		
Initial weight, kg	16.77	17.00	16.77	0.12	0.694
Final weight, kg	48.75 ^b^	49.75 ^b^	54.94 ^a^	0.79	<0.001
ADG, kg/d	0.55 ^b^	0.56 ^b^	0.66 ^a^	0.01	<0.001
ADFI, kg/d	1.43 ^b^	1.48 ^b^	1.57 ^a^	0.02	0.004
FCR	2.60 ^ab^	2.62 ^b^	2.39 ^a^	0.04	0.031

Dry = solid feed group; Liq = liquid feed group; Pro = probiotic-enriched liquid feed group; ADG represents average daily weight gain; ADFI represents average daily feed intake; FCR represents feed conversion ratio. ^a,b^ Values within the same row labeled with different superscript letters are significantly different (*p* < 0.05). All data are presented as mean and SEM.

**Table 2 ijms-26-05698-t002:** Effects of different feeding regimes on nutrient digestibility in nursery piglets.

Item, %	Group			SEM	*p*-Value
	Dry	Liq	Pro		
Dry matter	84.61	84.73	85.96	0.31	0.148
Gross energy	85.09	85.24	86.35	0.33	0.247
Crude protein	86.11	86.88	87.03	0.38	0.639
Ether extract	63.93	66.56	70.76	1.33	0.085
Total amino acid	88.89	89.73	90.08	0.35	0.419

Dry = solid feed group; Liq = liquid feed group; Pro = probiotic-enriched liquid feed group. All data are presented as mean and SEM.

**Table 3 ijms-26-05698-t003:** Ingredients and chemical composition of basal diet (as-fed basis) ^1^.

Item	Phase (kg)	
	15–25	25–50
Ingredients, %		
Corn	65.60	67.53
Rice bran	2.00	3.00
Puffed soybean	10.00	8.00
Soybean meal	15.00	18.00
Soybean oil	2.00	0.00
Fish meal	3.00	1.00
Salt	0.30	0.30
Limestone	0.96	1.00
CaHPO_4_	0.40	0.47
premix ^2^	0.20	0.20
L-Lysine	0.30	0.31
L-Threonine	0.12	0.09
DL-Methionine	0.12	0.10
Total	100.00	100.00
Measured nutrient level		
Gross energy, kcal/100 g	394.20	386.20
Dry matter, %	87.70	87.90
Crude protein, %	18.77	18.53
Ether extract, %	5.70	4.10
Calcium, %	0.74	0.76
Total phosphorus, %	0.50	0.48
Lysine, %	1.26	1.27
Methionine, %	0.37	0.40
Threonine, %	0.86	0.80

^1^ Basal diet formulated did not include any additional microecological preparations. ^2^ Each kilogram of the complete dietary formulation supplied following: VA, 9000 IU; VD3, 3000 IU; VE, 20 IU; VK3, 3.0 mg; VB1, 1.5 mg; VB2, 4.0 mg; VB6, 3.0 mg; VB12, 0.2 mg; nicotinic acid, 30 mg; D-pantothenic acid, 15 mg; folic acid, 0.75 mg; biotin, 0.1 mg; Fe, 100 mg; Cu, 6 mg; Zn, 100 mg; Mn, 4.0 mg; I, 0.14 mg; Se, 0.3 mg.

**Table 4 ijms-26-05698-t004:** Primers used for quantitative real-time PCR.

Genes	Forward Sequence (5′→3′)	Reverse Sequence (5′→3′)
*GAPDH*	ACACTCACTCTTCCACTTTTG	CAAATTCATTGTCGTACCAG
*Mucin2*	GACTACAACTTCGCCTCCGA	GACCGTCAGCAGGATGTACT
*ZO-1*	ATCTCGGAAAAGTGCCAGGA	CCTTCCCCTCAGAAACCCAT
*Occludin*	CAGGTGCACCCTCCAGATTG	ATGTCGTTGCTGGGTGCATA
*Claudin1*	TACTTTCCTGCTCCTGTC	AAGGCGTTAATGTCAATC

## Data Availability

The raw sequencing reads of colon intestinal microbes in pigs were deposited into the NCBI Sequence Read Archive (SRA) database (Accession Number: PRJCA023433).

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
