# Peer review of "Composite Probiotics Improve Gut Health and Enhance Tryptophan Metabolism in Nursery Piglets During Liquid Feeding"

_ijms, 2025, doi:10.3390/ijms26125698_

Round 1
Reviewer 1 Report
Comments and Suggestions for Authors
This study investigated the synergistic effects of a probiotic consortium delivered via a liquid feeding system on gut health, microbial ecology, and tryptophan metabolism in nursery piglets. The integration of multi-omics approaches (16S rRNA sequencing, metabolomics, and targeted metabolite analysis) provides a comprehensive understanding of the mechanisms underlying probiotic efficacy. The work is well-structured, innovative, and addresses a relevant topic in livestock nutrition. However, several aspects require clarification or expansion to strengthen the scientific rigor and practical applicability of the findings. Below are my specific comments:
Major Comments:
Q1: The study administered a probiotic consortium (500 g/t of feed) containing Bifidobacterium infantis, Lactobacillus plantarum, and Pediococcus acidilactici (2:1:1 ratio). However, the rationale for selecting this specific dosage, strain combination, and ratio is not explicitly justified. Were these parameters based on prior in vitro studies, industry standards, or previous in vivo trials in swine? Clarification is needed to ensure reproducibility of the results.
Q2: The manuscript does not specify the timing of sample collection (e.g., serum, colonic contents) relative to feeding schedules or circadian rhythms. Diurnal fluctuations in gut microbiota composition and metabolite levels are well-documented. How were these variables controlled? For instance, were samples collected at a standardized time post-feeding to minimize variability?
Q3: While the Pro group demonstrated significant improvements in growth performance and gut health, the economic feasibility of probiotic-enriched liquid feeding in commercial swine production remains unclear. What is the estimated cost-benefit ratio of this intervention? A brief discussion of scalability, shelf-life challenges in liquid systems, and compatibility with industrial feed processing would enhance practical relevance.
Minor Comments:
- Table 1: Replace “F: G” with “FCR” (feed conversion ratio) for clarity and alignment with standard terminology.
- B. infantis and L. plantarum should be italicized (e.g., line 515). Ensure taxonomic names are formatted correctly throughout the manuscript.
- Table 2: Include probiotic CFU/g values in the main text (currently in Methods). Also, revise the table title to “Ingredients and chemical composition of diets.”
- Specify the exact procedure for adding probiotics to the liquid feed, including timing (e.g., immediately before feeding or earlier) and duration of exposure prior to administration.
- Line 379: Clarify the intestinal sampling site, as microbial composition varies significantly along the gastrointestinal tract.
- The in vitro assay describing probiotic-derived ILA production from tryptophan lacks detail. Please elaborate on the experimental setup, detection method, and metabolite quantification procedures.
- Liquid Feeding System Limitations: Briefly acknowledge potential limitations of liquid feeding (e.g., nutrient sedimentation, microbial contamination risks) in the Discussion to contextualize the practicality of probiotic delivery.
- Condense the speculative mechanisms on ILA (e.g., AhR/NF-κB) to focus on observed correlations.
- Ensure consistency and completeness across all references, particularly with regard to journal abbreviations and page numbers.
- Correct all instances of unitalicized variables (e.g., lowercase n for sample size) and ensure proper formatting throughout the manuscript.
- Please check the full text for grammar and spelling.
Author Response
Thank you for your kind guidance. As our revisions involve formatting, reference consistency, and other layout-related details, we have provided our point-by-point responses to the reviewers' comments in a Word document for clarity. Please refer to the uploaded file titled “Replies to Reviewer 1” for the complete response.

Reviewer 2 Report
Comments and Suggestions for Authors
Dear authors ,
I would like to offer several observations and suggestions for manuscript improvement s where further clarification is needed, from my point o view
The introduction points very well, the probiotics benefits to the specific context of nursery piglets and liquid feeding. The final paragraph clearly states the research goal and the methodological approach.
Rows 40 -43 The phrase is too long. Please break it into two sentences for clarity.
Please emphasize more the physiological challenges during the nursery phase and the mechanistic role of probiotics in this phase .
Why is Materials and Methods on chapter 4, usually goes as Chapter 2??? Please change it !
Please add the protocol numbers and the date of the experiment?!
Microclimate conditions please mention within Material and method chapter!
Rows 264-265 Only statistically significant results should be reported, as referencing slight increases or non-significant trends may lead to misinterpretation or overestimation of effects
Although nutrient composition and apparent total tract digestibility were assessed, the manuscript does not include any related results or discussion. Please provide detailed information on the nutrient profiles of the diets and the corresponding digestibility data.
Author Response
Thank you for your kind guidance. As our revisions involve formatting, reference consistency, and other layout-related details, we have provided our point-by-point responses to the reviewers' comments in a Word document for clarity. Please refer to the uploaded file titled “Replies to Reviewer 2” for the complete response.

Reviewer 3 Report
Comments and Suggestions for Authors
- Summary and General Assessment
This manuscript investigates the synergistic effect of composite probiotics (Bifidobacterium infantis, Lactobacillus plantarum, and Pediococcus acidilactici) incorporated into a liquid feeding system on growth performance, gut health, and tryptophan metabolism in nursery piglets. The study is comprehensive and employs a multi-omics approach—microbiome profiling (16S rRNA), untargeted and targeted metabolomics, gene expression, and in vitro validation—to uncover underlying mechanisms.
Overall impression:
The manuscript is well-conceived, methodologically sound, and the data are rich and well-analyzed. It addresses a timely and relevant topic in swine nutrition and microbiome interaction, with solid implications for animal health and productivity.
- Major Strengths
- Innovative design: The integration of probiotic delivery within a liquid feeding system is novel and practically relevant for modern swine production.
- Multi-omics approach: The combination of microbiome sequencing, metabolomics, gene expression, and in vitro confirmation strengthens mechanistic interpretations.
- Clarity and coherence: The manuscript is logically structured and clearly written, with appropriate use of visuals and statistical analyses.
- Significant findings:
- Improved growth metrics and gut integrity in the Pro group.
- Modulation of microbial composition (increase in beneficial genera).
- Activation of tryptophan metabolic pathways, particularly the kynurenine and indole-3-lactic acid pathways.
- Strong correlation between metabolite profiles and microbial taxa.
- Major Concerns and Suggestions
3.1. Justification of Probiotic Strains
- While the inclusion of B. infantis and L. plantarum is well justified, the manuscript lacks rationale for including Pediococcus acidilactici—especially since it was not detected in vivo. Authors should briefly discuss why this strain was included and hypothesize why it may have failed to colonize.
3.2. Mechanistic Claims
- Although the association between probiotics and tryptophan metabolism is well supported, causality remains speculative. While in vitro assays support the metabolic capacity of the strains, it’s unclear how these pathways were activated in vivo (e.g., whether microbial or host-derived IDO1 played a dominant role). Consider a more cautious tone in the Discussion.
3.3. Liquid Feed Composition
- Clarify whether the same basal diet was used for all groups, including the solid and liquid diets, and whether particle size or nutrient distribution changed upon mixing. Differences in digestibility could stem from matrix effects rather than probiotics.
3.4. Microbiota Statistical Interpretation
- While PERMANOVA and MaAsLin2 are appropriate, variation explained by treatments is relatively low (6–12%). Consider including an additional discussion on inter-animal variation and whether observed shifts are biologically meaningful despite modest R² values.
- Minor Comments
- Abstract: Consider adding quantitative values (e.g., % increase in ADG or SCFA levels) for better impact.
- Figure legends: Ensure all abbreviations (e.g., DAO, iFABP) are defined within figure legends for standalone clarity.
- Reference to methodology: The manuscript mentions a 60-day trial but does not specify the weaning age or the age range of piglets—please clarify.
- Gene expression: Only mRNA expression is evaluated—any plans to validate protein expression via Western blot or immunohistochemistry?
- Discussion: Could the enhanced L-tryptophan absorption be confirmed via transporter expression (e.g., SLC6A19)?
- Language: Mostly clear, but a few sentences could benefit from editing for conciseness (e.g., lines 265–269, 296–298).
- Recommendation
Recommendation: Minor Revision
The manuscript is of high quality and provides valuable insights into the role of probiotics in gut health and metabolism. Addressing the comments above—particularly clarifying rationale for strain inclusion, adding nuance to mechanistic claims, and ensuring methodological transparency—will strengthen the manuscript further.
Author Response
Thank you for your kind guidance. As our revisions involve formatting, reference consistency, and other layout-related details, we have provided our point-by-point responses to the reviewers' comments in a Word document for clarity. Please refer to the uploaded file titled “Replies to Reviewer 3” for the complete response.

Round 2
Reviewer 2 Report
Comments and Suggestions for Authors
Dear authors,
I am pleased to see that the manuscript has been significantly improved.
Table 2 are expressed as percentages (%), therefore you should add it into the table.